# Macrophage and dendritic cell subset composition can distinguish endotypes in adjuvant-induced asthma mouse models

**Müge Özkan**[1], **Yusuf Cem Eskiocak**[2], **Gerhard Wingender**[2,3]*

1 Department of Genome Sciences and Molecular Biotechnology, Izmir International Biomedicine and Genome Institute, Dokuz Eylul University, Balcova/Izmir, Turkey, 2 Izmir Biomedicine and Genome Center (IBG), Balcova/Izmir, Turkey, 3 Department of Biomedicine and Health Technologies, Izmir International Biomedicine and Genome Institute, Dokuz Eylul University, Balcova/Izmir, Turkey

* gerhard.wingender@ibg.edu.tr

**Citation:** Özkan M, Eskiocak YC, Wingender G (2021) Macrophage and dendritic cell subset composition can distinguish endotypes in adjuvant-induced asthma mouse models. PLoS ONE 16(6): e0250533. https://doi.org/10.1371/journal.pone.0250533

**Data Availability Statement:** All relevant data are within the paper and its Supporting information files.

## Abstract

Asthma is a heterogeneous disease with neutrophilic and eosinophilic asthma as the main endotypes that are distinguished according to the cells recruited to the airways and the related pathology. Eosinophilic asthma is the treatment-responsive endotype, which is mainly associated with allergic asthma. Neutrophilic asthma is a treatment-resistant endotype, affecting 5–10% of asthmatics. Although eosinophilic asthma is well-studied, a clear understanding of the endotypes is essential to devise effective diagnosis and treatment approaches for neutrophilic asthma. To this end, we directly compared adjuvant-induced mouse models of neutrophilic (CFA/OVA) and eosinophilic (Alum/OVA) asthma side-by-side. The immune response in the inflamed lung was analyzed by multi-parametric flow cytometry and immunofluorescence. We found that eosinophilic asthma was characterized by a preferential recruitment of interstitial macrophages and myeloid dendritic cells, whereas in neutrophilic asthma plasmacytoid dendritic cells, exudate macrophages, and GL7+ activated B cells predominated. This differential distribution of macrophage and dendritic cell subsets reveals important aspects of the pathophysiology of asthma and holds the promise to be used as biomarkers to diagnose asthma endotypes.

## Introduction

Asthma is a chronic airway inflammation with often debilitating impacts on the health of patients. Based on the immune cells that infiltrate into the lung, several subtypes or endotypes for asthma have been described [1–3]. Type 2-high or eosinophilic asthma is the best-understood endotype and is often triggered by inhaled antigens, like house dust mites (HDM), ragweed pollen, mold, or cockroach proteins [1, 4]. This eosinophilic asthma is characterized by (i) elevated levels of eosinophils (eosinophilia) and mast cells in the bronchia, by (ii) a type 2-polarized immune response, with the production of Th2 cytokines, like IL-4, IL-5, and IL-13, and by (iii) an augmented production of antigen-specific IgE antibodies [1, 5, 6]. About half of all asthma patients have such an eosinophilic airway inflammation [1, 7] and

**Funding:** This work was funded by grants from TUBITAK (116Z272, GW), the Dokuz Eylul University (2016.KB.SAG.020 and 2017.KB.SAG.029, GW), and the European Molecular Biology Organization (EMBO, Installation Grant 3073; GW). The funders had no role in study design, data collection and analysis, decision to publish, or preparation of the manuscript.

**Competing interests:** The authors have declared that no competing interests exist.

**Abbreviations:** AM, alveolar macrophage; BALF, bronchoalveolar lavage fluid; CFA, complete Freud's adjuvant; COPD, chronic obstructive pulmonary disease; DC, dendritic cell; ExM, exudate macrophage; H&E, hematoxylin-eosin; HDM, house dust mite; iBALT, inducible bronchus-associated lymphoid tissue; IM, interstitial macrophage; mDC, myeloid dendritic cell; OVA, ovalbumin; PAS, periodic acid-Schiff stain; pDC, plasmacytoid dendritic cell.

corticosteroids, anti-IgE, anti-IL-5/IL-5R, and anti-IL-4/IL-13 treatment are effective at alleviating the symptoms [1, 8, 9]. However, such treatment is less effective in patients with a type 2-low, neutrophil-dominated form of asthma [10, 11]. This neutrophilic asthma is characterized, besides the neutrophilia, by an elevated Th1/17 immune response, indicated by the cytokines IFNγ and IL-17 [1, 12–16]. Although treatments of type-2 low asthma by blocking IL-17, IL1β, or CXCR2 are in clinical trials, reports show that only a portion of the patients benefitted, suggesting the etiology of neutrophilic asthma is more complex [17].

Several *in vivo* mouse models were developed to study the pathogenic mechanisms underlying the endotypes and to develop endotype-specific treatment strategies. These mouse models differ in the means by which lung inflammation is induced. Asthma-like eosinophilic airway inflammation can be modeled by adjuvant-driven immunization [15, 18–21], chronic antigen administration [19, 22], and transfer of antigen-pulsed cells [18, 19, 23]. Asthma-like neutrophilic airway inflammation can be induced in mice by subcutaneous administration of either CFA [21, 24], LPS [14, 25], or poly(I:C) [26], along with a model antigen, like ovalbumin (OVA). For severe non-type 2 asthma, there is still a significant gap between disease-associated pathological features and a particular clinical outcome or treatment strategies. Hence, further research into immunological features and molecular mechanisms is required to develop new endotype-specific asthma treatments.

To determine additional immunophenotypic features specific for asthma-like neutrophilic airway inflammation, we adopted here mouse models that allowed for a direct side-by-side comparison of asthma-like eosinophilic and neutrophilic lung inflammation [15, 20, 21]. Specifically, we compared CFA/OVA-induced neutrophilic and Alum/OVA-induced eosinophilic asthma endotypes in mice [15, 20, 21] and provide an in-depth immunological analysis of the inflamed lung. Our data show that the two asthma endotypes demonstrate a significantly different distribution of macrophage and dendritic cell subsets.

## Results

### Adjuvant-induced mouse models of asthma endotypes

To study the immune response in the inflamed lung during eosinophilic and neutrophilic asthma side-by-side, we adopted two adjuvant-induced mouse models [15, 21] (Fig 1A). We first confirmed that the models faithfully reflect the eosinophilic and neutrophilic asthma endotypes. As expected, the cell analysis from BALF and lung showed high levels of eosinophils (Fig 1B, S1A Fig) or neutrophils (Fig 1C, S1B Fig) in the eosinophilic (Alum/OVA) or neutrophilic (CFA/OVA) asthma groups, respectively. In both cases, the data from the BALF mirrored the data obtained from the lung (Fig 1B and 1C). Immunohistochemical staining of the inflamed lungs showed severe inflammation with bleeding in the neutrophilic (CFA/OVA) group (Fig 1D, upper panel). However, the hyper-mucus secretion, which is an important driver of the airway obstruction in asthma [1], was comparable for both endotypes (Fig 1D, lower panel). Furthermore, in line with the literature [27, 28], the Th2 cytokines IL-5 and IL-13 were prevalent in eosinophilic asthma, whereas the Th1 cytokine IFNγ predominated in neutrophilic asthma (Fig 1E). Next, the antigen-specific humoral immune response was quantified. For serum IgE, higher levels were reported in eosinophilic asthma patients than in neutrophilic asthma patients [29, 30], and αIgE therapy was effective for eosinophilic asthma [29, 31]. However, in our mouse models, the OVA-specific serum IgE antibody levels were comparable in the eosinophilic (Alum/OVA) and neutrophilic (CFA/OVA) asthma groups (Fig 1F). Furthermore, and in line with clinical data [32, 33], the antigen-specific serum IgG and IgG1 antibody levels were significantly higher in the eosinophilic than the neutrophilic asthma endotype (Fig 1F). No difference was observed for OVA-specific IgG2b and IgG2c antibody

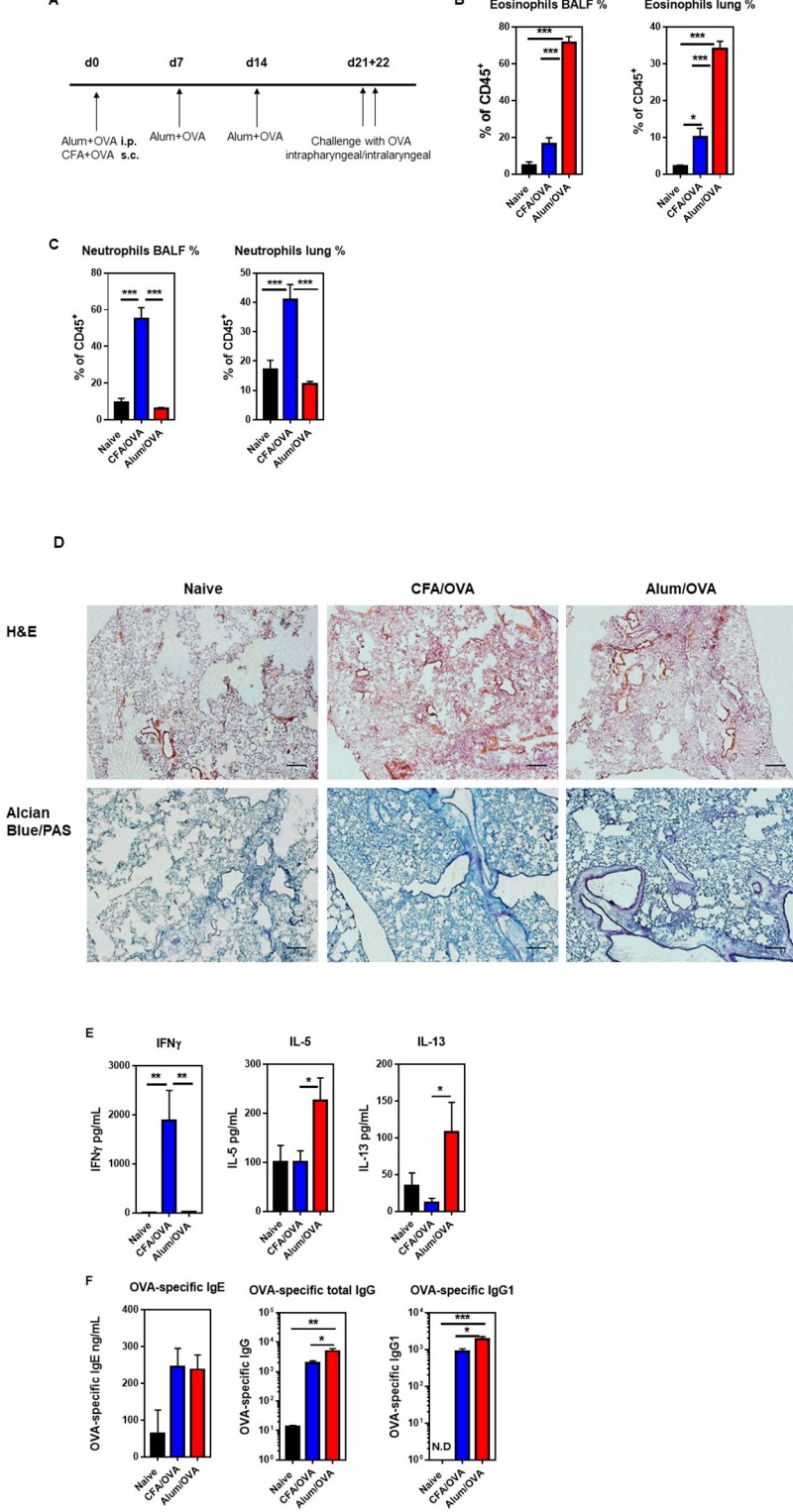

**Fig 1. Neutrophilic and eosinophilic asthma can successfully be modeled in mice. (A)** Experimental design and immunization timeline. C57BL/6 mice were either immunized three times weekly intraperitoneal (i.p.) with Alum (1 mg/mouse) for eosinophilic asthma or once subcutaneous (s.c.) with CFA (0.5 mg/mL) for neutrophilic asthma along with the model antigen OVA (20 μg/mouse). Three weeks after the first administration, the experimental groups were challenged with OVA (50 μg/mouse) daily for two days. Samples were collected 16–18 hours post-challenge. Cells from

bronchoalveolar lavage fluid (BALF) and lung homogenates were stained for **(B)** eosinophils (live CD45$^+$ CD19$^-$ CD11c$^-$ CD11b$^-$ Ly6G$^-$ Siglec-F$^+$) and **(C)** neutrophils (live CD45$^+$ CD19$^-$ CD11b$^{+/lo}$ Ly6G$^+$). The graphs showing the absolute cell counts of eosinophils and neutrophils in the BALF are shown in S1A and S1B Fig, respectively. **(D)** H&E (upper panel) and Alcian Blue/PAS (lower panel) immunohistochemistry analysis of inflamed lungs collected from indicated groups (10X magnification, scale 127 μm). **(E)** ELISA values of IFNγ, IL-5, and IL-13 from indicated BALF supernatants. **(F)** OVA-specific IgE, total IgG, and IgG1 ELISA results in serum indicated groups. For B, C, and F, combined data from four independent experiments are shown with 6 (F) or 12 (B, C) mice in the naïve group and 16 (F) or 18–19 (B, C) mice in the experimental asthma groups in total. For E, combined data from three independent experiments are shown with n = 9 mice in the naïve and n = 14–15 mice in experimental groups. The gating strategy for the myeloid cells is detailed in the S2 Fig.

levels (S1C Fig). These data demonstrate that the mouse models used here successfully replicate key features of human eosinophilic and neutrophilic asthma and are suitable to investigate asthma endotypes.

## Neutrophilic asthma is characterized by high numbers of plasmacytoid dendritic cells in the BALF

We next optimized a 19-parameter flow cytometry panel to quantify myeloid cells in addition to eosinophils and neutrophils, including basophils, mast cells, monocytes, as well as dendritic cells, and macrophage subsets in the bronchoalveolar lavage fluid (BALF) and lung samples (S2 Fig). A more robust influx of leukocytes into the BALF was observed following neutrophilic (CFA/OVA) than eosinophilic (Alum/OVA) asthma, with up to 4 x 10$^5$ total cells/mL (Fig 2A). The myeloid cell populations recovered from BALF, sorted from the highest to the lowest frequency, were neutrophils/eosinophils (Fig 1B and 1C), monocytes, macrophages, dendritic cells (DCs) (Fig 2A), and few basophils (Fig 2B) and mast cells (Fig 2C). In contrast to the BALF, basophils (Fig 2B), and mast cells (Fig 2C) were found to be enriched in the lung tissue of mice with eosinophilic (Alum/OVA) asthma. These data indicate that basophils and mast cells do not migrate to the BALF during lung inflammation. Although DCs constituted less than 1% of the BALF cells (Fig 2A), they were previously associated with the induction of Th2 immune responses in patients with allergic asthma [34]. To better understand the movement of DCs and their role in eosinophilic vs. neutrophilic asthma, we determined the frequency of DC subsets in the inflamed lung as CD11b$^+$ conventional (myeloid) DC (cDCs), CD103$^+$ lung resident DCs, and CD45R$^+$ plasmacytoid dendritic cells (pDCs). Both CD11b$^+$ cDC (Fig 2D) and lung resident CD103$^+$ DCs (Fig 2E) were more frequent in the lung tissue of mice with eosinophilic (Alum/OVA) than neutrophilic (CFA/OVA) asthma. In contrast, the frequency of pDCs did not change in the lung during the inflammation (Fig 2F). However, pDCs appeared to be the only DC population to migrate to the alveolar space, which was more prominent during neutrophilic (CFA/OVA) than eosinophilic (Alum/OVA) asthma (Fig 2F).

**The frequency of exudate and interstitial macrophages markedly differs between asthma endotypes.** In the steady-state, pulmonary macrophages consist mainly of two subsets, alveolar macrophages (AMs) and interstitial macrophages (IMs) [35]. AMs, which represent the major portion of lung resident macrophages, are tissue-resident macrophages of embryonic origin that self-renew in the lung [36, 37]. In contrast, IMs are derived from blood monocytes that infiltrated into the lung parenchyma [37, 38]. During lung inflammations, another blood monocyte-derived macrophage population that can be found in the lung are the non-resident Ly6C$^{hi}$ or exudate macrophages (ExMs) [39–42]. To characterize the turnover of macrophages in the two asthma endotypes, the frequency of AMs, IMs, and ExMs was determined in the BALF and lungs. In line with the large increase of macrophages in the lung (Fig 2A), the frequency of AMs considerably declined in the BALF and the lung of both asthma

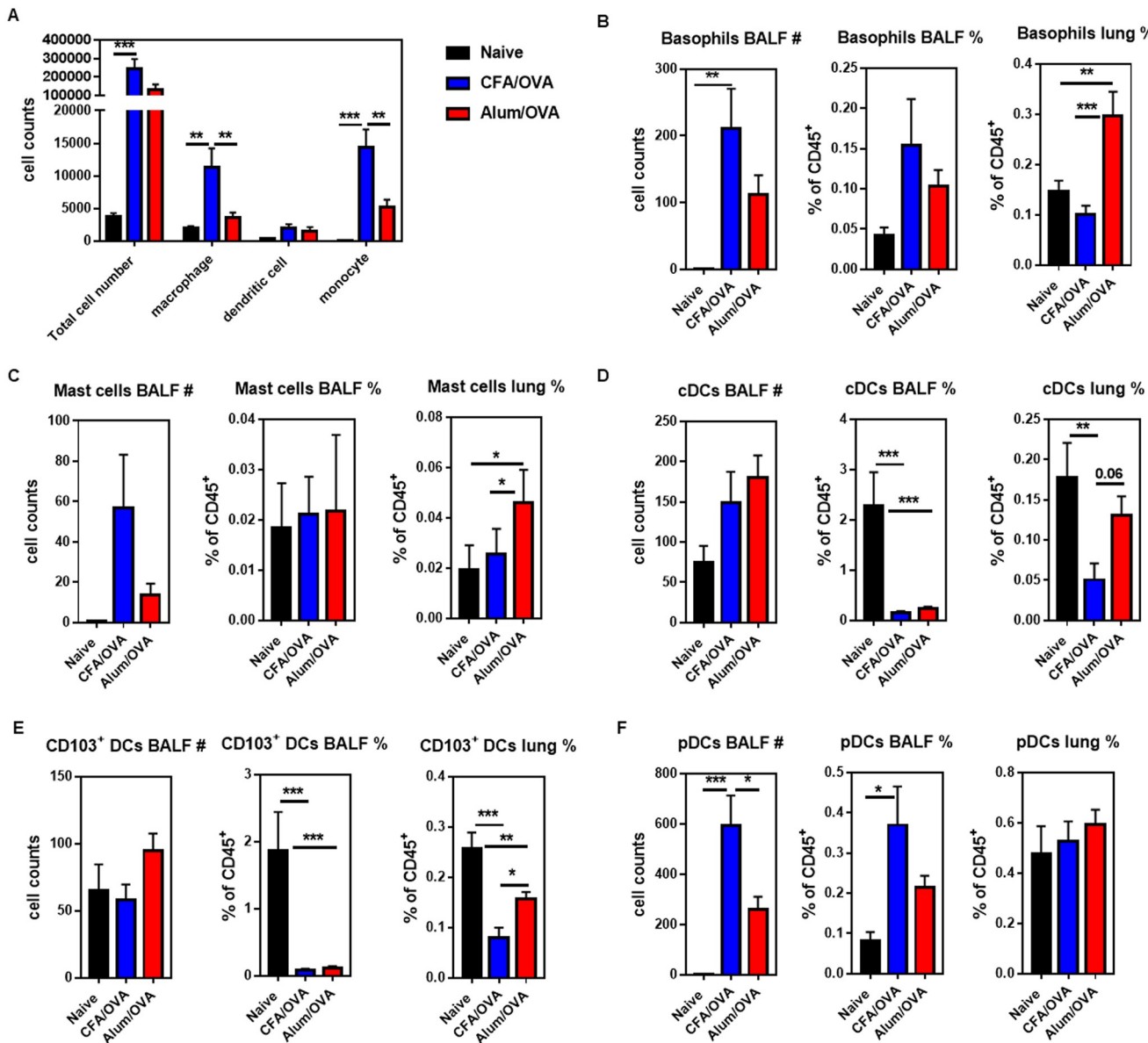

**Fig 2. DC subsets, mast cells, and basophils are differentially recruited to the inflamed lungs in neutrophilic and eosinophilic asthma.** C57BL/6 mice were immunized as outlined in Fig 1A to induce neutrophilic (CFA/OVA) or eosinophilic (Alum/OVA) asthma. Cells from bronchoalveolar lavage fluid (BALF) and lung homogenates were analyzed for indicated cell populations. **(A)** Total cell counts of all leukocytes (live CD45⁺ cells), macrophages (live CD19⁻ CD45⁺ Siglec-F⁻ Ly6G⁻ F4/80⁺ CD64⁺), dendritic cells (live CD19⁻ CD45⁺ Siglec-F⁻ Ly6G⁻ F4/80⁻ CD64⁻ CD24⁺ CD11c⁺ MHC class II⁺/⁻), and monocytes (live CD19⁻ CD45⁺ Siglec-F⁻ Ly6G⁻ F4/80⁻ CD64⁻ CD24⁻ CD11c⁻ MHC class II⁺/⁻ Ly6C⁺) in the BALF. Total cell counts and relative cell frequencies of **(B)** basophils (live CD45⁺ CD19⁻ Siglec-F⁻ Ly6G⁻ CD11b⁺ FcεRIα⁺ CD117⁻ cells); **(C)** mast cells (live CD45⁺ CD19⁻ Siglec-F⁻ Ly6G⁻ CD11b⁺ FcεRIα⁺ CD117⁺ cells); **(D)** conventional DCs (cDC, live CD45⁺ CD19⁻ Siglec-F⁻ Ly6G⁻ CD24⁺ CD11c⁺ MHC class II⁺ CD103⁻ CD11b⁺ cells); **(E)** CD103⁺ DCs (live CD45⁺ CD19⁻ Siglec-F⁻ Ly6G⁻ CD24⁺ CD11c⁺ MHC class II⁺ CD103⁺ cells); and **(F)** plasmacytoid dendritic cells (pDC, live CD19⁻ CD45⁺ Siglec-F⁻ Ly6G⁻ CD11b⁻ CD45R⁺ Ly6C⁺) in indicated organs. Combined data from four (A, D-F) or three (B, C) independent experiments are shown with 3 mice per naïve group and 4–5 mice per experimental asthma group in each experiment.

groups compared to the control group (Fig 3A). As IMs are located mainly in the lung parenchyma [37], it was expected that their numbers did not greatly increase in the BALF, leading to a sharp decline in their relative frequency in the BALF (Fig 3B). Interestingly, however, an increase in the frequency of IMs in the lung was only observed in the mice with eosinophilic

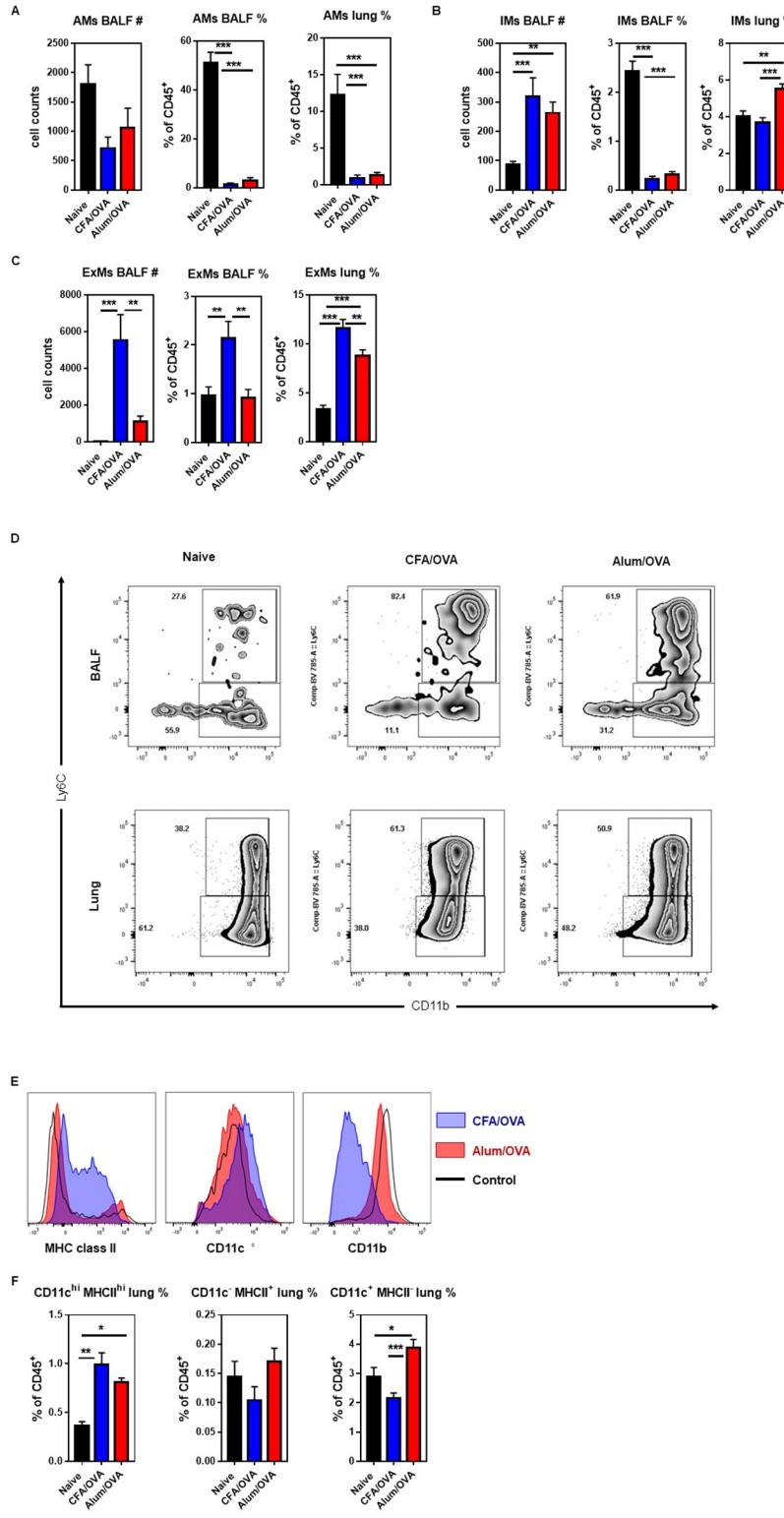

**Fig 3. The distribution of lung macrophage subsets among asthma endotypes.** C57BL/6 mice were immunized as outlined in Fig 1A to induce neutrophilic (CFA/OVA) or eosinophilic (Alum/OVA) asthma. Cells from bronchoalveolar lavage fluid (BALF) and lung homogenates were analyzed for indicated cell populations: **(A)** alveolar macrophages (AMs, live CD45$^+$ CD19$^-$ Siglec-F$^+$ Ly6G$^-$ CD11c$^+$ cells), **(B)** interstitial macrophages (IMs, live CD45$^+$ CD19$^-$ Siglec-F$^-$ Ly6G$^-$ CD24$^-$ F4/80$^+$ CD64$^{+/-}$ Ly6C$^-$ CD11b$^+$ cells); and **(C)** exudate macrophages (ExMs, live CD45$^+$

CD19$^-$ Siglec-F$^-$ Ly6G$^-$ CD24$^-$ F4/80$^+$ CD64$^{+/-}$ Ly6C$^+$ CD11b$^+$ cells). **(D)** Representative dot plots showing IMs (Ly6C$^-$ CD11b$^+$ cells) and ExMs (Ly6C$^+$ CD11b$^+$ cells) in BALF (upper panel) and lung (lower panel) from indicated groups. **(E)** Representative histograms showing the expression of MHC class II, CD11b, and CD11c on bronchial IMs from indicated groups. **(F)** Bronchial IMs (BIMs) were subdivided according to their expression of CD11c and MHC class II and the frequencies of BIM1 (CD11c$^+$ MHC class II$^+$, left panel), BIM2 (CD11c$^-$ MHC class II$^+$, middle panel), and BIM3 (CD11c$^+$ MHC class II$^{neg}$, right panel) cells are shown. Combined data from four independent experiments are shown (naïve group n = 12 mice/group, experimental asthma groups n = 17–18 mice/group in total).

(Alum/OVA) and not neutrophilic (CFA/OVA) asthma (Fig 3B). In contrast, the influx of ExMs was more prominent in the neutrophilic (CFA/OVA) than the eosinophilic (Alum/OVA) asthma group, which was particularly striking in the BALF (Fig 3C). Recently, it was suggested that the IMs themselves can be divided into three subsets, based on the expression of CD11b, CD11c, and MHC class II [38]. When these subsets were analyzed (Fig 3D–3F), we noticed that the IM population in the eosinophilic (Alum/OVA) inflammation mainly consisted of the CD11c$^+$ MHC class II$^{neg}$ population (Fig 3E and 3F), which was described to have the highest phagocytic activity and the lowest turnover rate among the IM subsets [38]. Together, these data indicate that eosinophilic and neutrophilic asthma involve significantly different macrophage populations, with IMs increasing in the lungs with eosinophilic asthma and ExMs increasing in neutrophilic asthma.

## Th17 cells, but not NKT17 cells, are significantly expanded in neutrophilic asthma

Both conventional CD4$^+$ T cells [43, 44] and innate-like invariant Natural Killer (iNKT) cells [45–48] have been shown to be involved in asthma in humans and mouse models. Therefore, we next measured the subset distributions of CD4$^+$ T cells (Th1, Th2, Th17, Treg) and iNKT cells (NKT1, NKT2, NKT17) in our eosinophilic and neutrophilic asthma models. Surprisingly, for the frequency of Th1 and Th2 cells and of Tregs, no significant difference was observed in the lungs of mice with eosinophilic (Alum/OVA) or neutrophilic (CFA/OVA) asthma (Fig 4A). Similarly, the production of the cytokines IFNγ, IL-4, IL-13, and TNF by CD4$^+$ T cells did not differ between the eosinophilic (Alum/OVA) and neutrophilic (CFA/OVA) asthma groups (Fig 4B). In contrast, the frequency of Th17 cells in the lung (Fig 4A), as well as the production of IL-17 by CD4$^+$ T cells (Fig 4B), was significantly higher in the lungs of mice with neutrophilic (CFA/OVA) than with eosinophilic (Alum/OVA) asthma. The lung iNKT cells (Fig 4C) were mainly NKT1 cells (Fig 4D) and their frequency increased during neutrophilic (CFA/OVA) asthma (Fig 4D). No changes were observed in the lungs for NKT2 cells (Fig 4D). However, the frequency of NKT17 cells decreased during neutrophilic (CFA/OVA) asthma (Fig 4D), which was, interestingly, the opposite of the changes observed for Th17 cells (Fig 4A).

## Neutrophilic asthma is characterized by an increase of activated B cell numbers

B cells, besides their ability to produce allergen-specific antibodies [49], are involved in the pathogenesis of chronic lung inflammation either by presenting antigens to T cells [50] or by the formation of 'inducible bronchus-associated lymphoid tissue' (iBALTs) [51]. iBALTs are tertiary lymphoid structures that are rapidly induced and provide an effective local site to drive lymphocyte activation and immune response in the lung [52]. To assess the B cell responses during eosinophilic (Alum/OVA) and neutrophilic (CFA/OVA) asthma, we measured the frequency of B cells and their cytokine production, as well as the frequency of germinal center

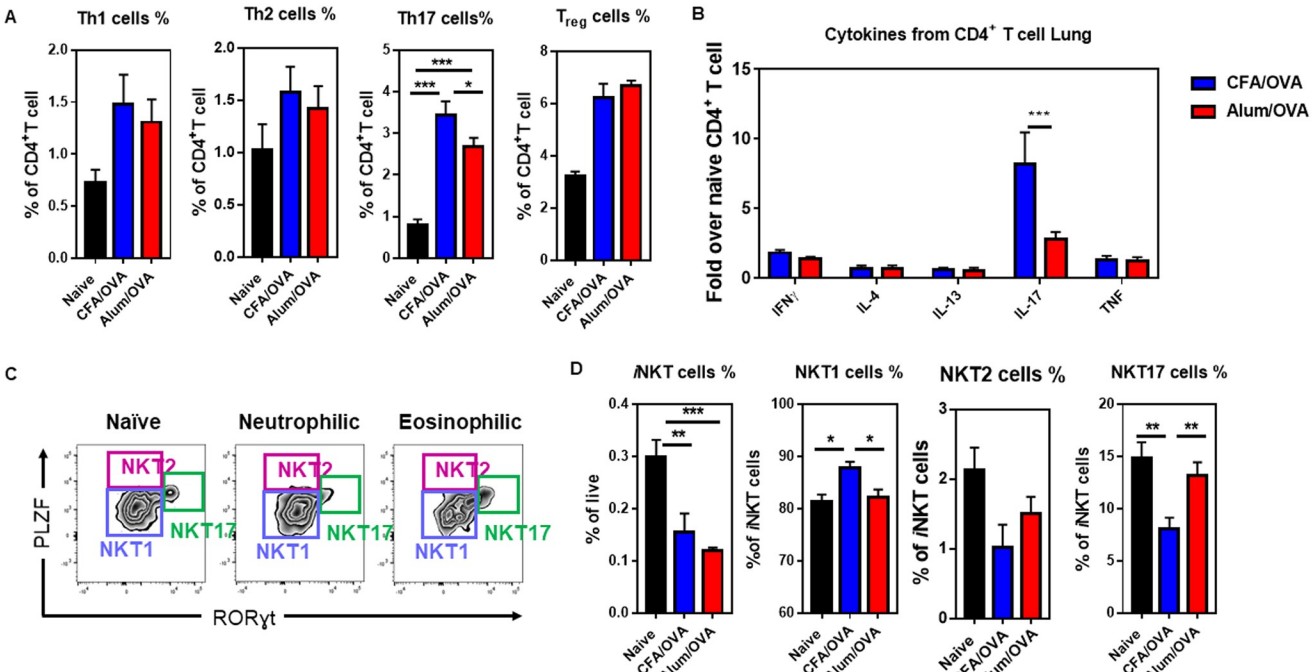

**Fig 4. Frequency of CD4+ T and _i_NKT cell subsets in the inflamed lungs during neutrophilic and eosinophilic asthma.** C57BL/6 mice were immunized as outlined in Fig 1A to induce neutrophilic (CFA/OVA) or eosinophilic (Alum/OVA) asthma. Cells from lung homogenates were stained for indicated cell populations. **(A)** The relative cell frequency of lung Th1 cells (live CD19- CD3ε+ CD4+ FoxP3- Tbet+ cells), Th2 cells (live CD19- CD3ε+ CD4+ FoxP3- Gata3+ cells), Th17 cells (live CD19- CD3ε+ CD4+ FoxP3- RORγt+ cells), and Tregs (live CD19- CD3ε+ CD4+ CD127lo/- FoxP3+ cells) is shown. **(B)** Production of the indicated cytokines by lung CD4+ T cells (live CD19- CD1d/PBS57-tetramer- CD3ε+ CD4+ cells) following _in vitro_ stimulation with PMA and ionomycin. **(C)** Gating for _i_NKT cell (live CD19- CD3ε+ CD1d/PBS57-tetramer+ cells) subsets in the lung (NKT1 cells: PLZFlo RORγt- cells, NKT2 cells: PLZFint/hi RORγt- cells, NKT17 cells: PLZFint RORγt+ cells). **(D)** Relative frequency of NKT1 and NKT17 cells in the lung of indicated mice. Combined data from three (C, D), four (A), or two (B) independent experiments are shown with 3 mice per naïve group and 4–5 mice per experimental asthma group in each experiment. The gating strategy for the lymphoid cells is detailed in the S3 Fig.

(GC) B cells in the lungs. Although the overall frequency of B cell was comparable between the BALFs and lungs with eosinophilic (Alum/OVA) or neutrophilic (CFA/OVA) asthma (Fig 5A), the IFNγ production of B cells from lungs with neutrophilic asthma was significantly higher than in the neutrophilic asthma group (Fig 5B). No difference was observed for IL-4, IL-17, and TNF production by the B cells (Fig 5B). Importantly, we noticed a significant increase of CD45R/B220+ GL7+ CD95+ germinal center (GC) B cells in the lungs of mice with neutrophilic (CFA/OVA) asthma, which was not observed in the lungs from mice with eosinophilic (Alum/OVA) asthma (Fig 5C). These data indicate that iBALT formation was only supported during neutrophilic asthma in our mouse models, a finding that was supported by the immunofluorescence of the inflamed lungs (Fig 5D).

## Discussion

Neutrophilic asthma responds poorly to the main treatment options currently available for asthma patients, severely impacting the quality of life of patients. A better understanding of the immunological features and underlying immunopathology of the different asthma endotypes is expected to lead the way to improved therapies. Here, we directly compared in detail the immune response (i) between asthma-like neutrophilic airway inflammation in mouse models and (ii) between BALF vs. lung tissue of those mice. Our data demonstrate that the

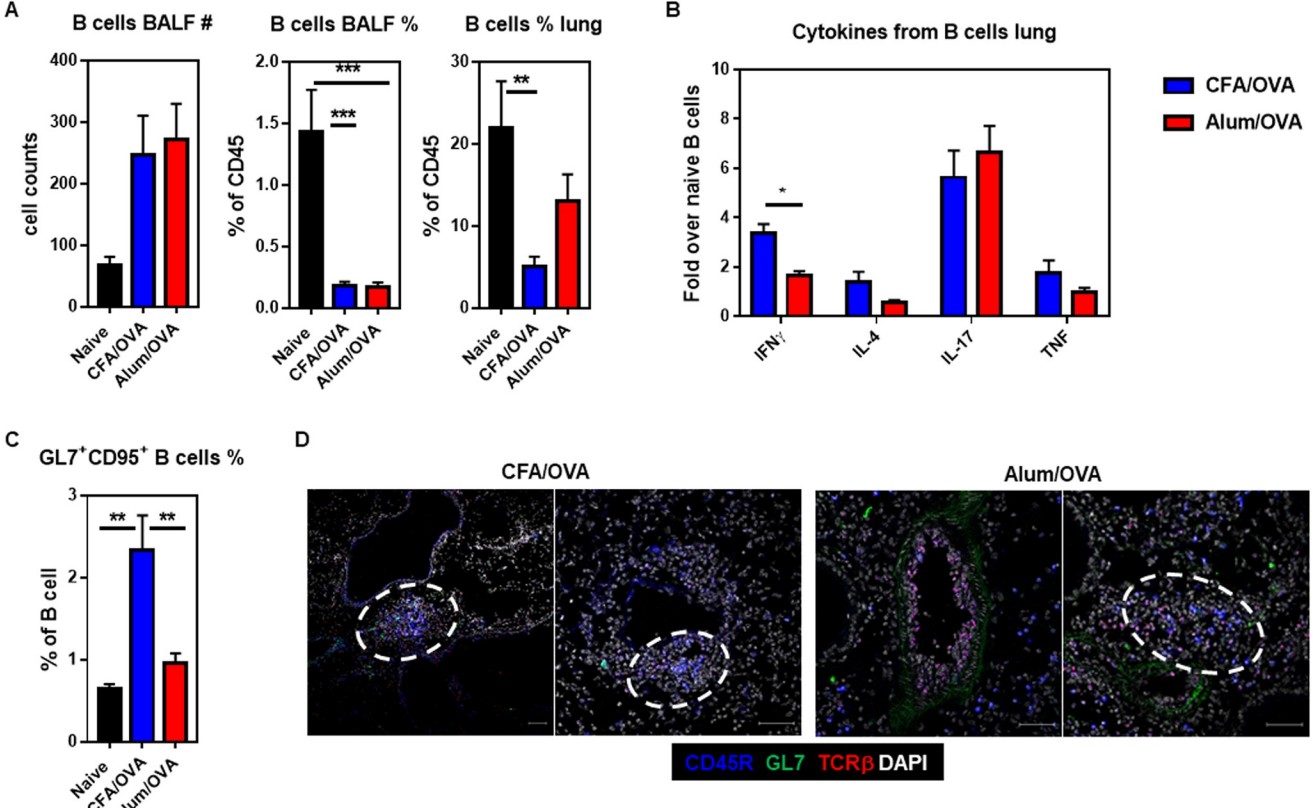

**Fig 5. iBALT formation is more pronounced in neutrophilic than in eosinophilic asthma.** C57BL/6 mice were immunized as outlined in Fig 1A to induce neutrophilic (CFA/OVA) or eosinophilic (Alum/OVA) asthma. Cells from bronchoalveolar lavage fluid (BALF) and lung homogenates were analyzed for indicated cell populations. **(A)** Total cell count (left panel) and relative cell frequencies of B cells (live CD45[+]) in indicated organs. **(B)** Production of the indicated cytokines by lung B cells (live CD45[+]) following *in vitro* stimulation with PMA and ionomycin. The values are given as fold-change of the two experimental groups over the control group. **(C)** Relative cell frequencies of germinal center (GC) B cells (live CD3ε[-] CD45R[+] CD95[+] GL7[+] cells) in lung homogenates. **(D)** Inflamed lungs were fixed with 4% PFA (4h, 4˚C), prepared for cryostat sectioning (7–10 μm), and stained with DAPI (white), CD45R-AF647 (blue), GL7-AF488 (green), and TCRβ-AF594 (red). Scale = 50 μm. Representative data from three biological replicates are shown. Unless indicated otherwise, combined data from two independent experiments are shown (naïve group n = 5 mice/group, experimental asthma groups n = 8–9 mice/group in total).

BALF provides a good immunological representation of the inflamed lung. Furthermore, asthma endotypes are characterized by the distinct distribution of several myeloid cells, besides eosinophils and neutrophils; namely cDCs, CD103[+] DCs, pDCs, interstitial macrophages (IMs), and exudate macrophages (ExMs). For these populations, several differences between BALF and lung tissue were noted, which indicates that pDCs and ExMs in BALF can identify neutrophilic asthma in mice. Furthermore, Th17 cells, but not NKT17 cells or germinal center B cells, were significantly expanded in neutrophilic asthma.

Asthma is a heterogeneous disease and new endotypes are being described in the clinic. However, many challenges remain. Airway inflammations, like asthma, have long been modeled *in vivo* in animals to gain insight into pathogenesis, progression, and treatment options. Despite their intrinsic limitations, such animal models are instrumental for our understanding of complex human diseases. Prime/boost immunization of mice with ovalbumin (OVA) together with the adjuvant Alum (aluminum hydroxide) is a well-described approach to induce a type-2 immune response in the airways, including antigen-specific IgEs, hypersensitivity responses, and airway remodeling [32]. Clinically relevant allergens, such as extracts of

house dust mites (HDM), have also been used in animal models [53, 54]. However, their use is hampered by technical difficulties and a lack of standardization, which reduces reproducibility [55]. For neutrophilic asthma, less work and fewer animal models have been reported [56]. Based on literature data and our preliminary results, we concluded that the adjuvant-induced eosinophilic (Alum/OVA) and neutrophilic (CFA/OVA) asthma models used here are best suited to compare asthma-like eosinophilic and neutrophilic lung inflammation side-by-side.

Basophils [57, 58] and mast cells [59, 60] are associated with allergen-induced airway inflammation due to IgE-mediated effector functions. In line with these reports, we noted a clear increase in basophils (Fig 2B) and mast cells (Fig 2C) in the lungs of mice with eosinophilic airway inflammation (Alum/OVA). Furthermore, our data indicate that basophils and mast cells do not migrate to the BALF during allergen-induced lung inflammation. To our knowledge, this is the first report showing this directly for eosinophilic and neutrophilic asthma. In Th2-low (neutrophilic) asthma, mast cells locate to the proximal airway epithelium [61, 62] and the submucosal region [63], whereas in Th2-high (eosinophilic) asthma, they are found in the intraepithelial region [61, 62]. Therefore, it seems possible that the location of the mast cells, and potentially the basophils, in the inflamed lung limits their propensity to migrate to the BALF.

In allergic, eosinophilic airway inflammation, the antigen presentation by pulmonary CD11b$^+$ cDCs and CD103$^+$ cDCs is essential for the induction of the Th2 response [64]. Although pDCs have the ability, similar to other pulmonary DCs, to take up and process antigens [65], they actually prevent the differentiation of effector T cells, and depletion of pDCs could exacerbate lung inflammation in an LPS-induced asthma model [66, 67]. In addition, pDCs are associated with the induction of type 1 immune responses and, for example, protect from viral bronchiolitis [68] and suppress ILC2 activity via IFNα during fungus-induced allergic asthma [69]. Similar to the literature, we found that conventional pulmonary DCs, including CD11b$^+$ and CD103$^+$ cDCs, are significantly expanded in eosinophilic asthma (Fig 2D and 2E). In contrast, in neutrophilic asthma, pDCs were the only DCs that infiltrated the alveolar space of the inflamed lung (Fig 2F). Although pDCs have been reported to increase upon allergen challenge in the BALF [70] and the induced sputum [71] of asthmatic patients, to our knowledge, no study compared their frequency in different endotypes. Interestingly, the increase of pDCs in the BALF was not accompanied by an increase of pDCs in the lung itself (Fig 2F).

Macrophages are important innate immune cells involved in tissue homeostasis and host defense. Lung macrophage subsets were initially defined by their location: alveolar macrophages (AMs) are mainly found in the alveolar lumen, whereas interstitial macrophages (IMs) reside in the lung interstitium [35, 72]. AMs are essential for the repair of lung injuries induced, e.g. by physical damage [73], LPS [74, 75], or infections [39]. Consequently, AMs are also protective during allergic airway inflammation [76] and lung fibrosis [77]. In contrast, IMs, which constitute approx. 9% of the macrophages in a healthy lung [38], have primarily proinflammatory functions. For example, they aggravate allergic inflammation [78] and fibrosis [79]. During lung inflammations, blood-monocytes can infiltrate the lung and give rise to a third lung macrophage population, Ly6C$^{hi}$ exudate macrophages (ExMs). ExMs are mainly proinflammatory and have been described, for example, during lung inflammations caused by diphtheria toxin [41] and infections with bacteria [39, 42], fungi [40], or viruses [80]. Due to their functional differences, it is important to analyze macrophages on the subset level during pulmonary diseases. When we analyzed AMs, IMs, and ExMs from the BALF and the inflamed lung of mice with eosinophilic and neutrophilic asthma, several changes were apparent. In line with previous findings [81], AMs significantly declined during both types of lung inflammation (Fig 3A). Importantly, we found that the frequency of IMs and ExMs distinguishes the

two asthma endotypes, with IMs being expanded more in eosinophilic asthma (Fig 3B) and ExMs being more prevalent in neutrophilic asthma (Fig 3C). Further analysis of the IMs based on surface markers [38], showed that the IMs in both endotypes mainly consisted of the CD11c$^{lo}$ MHC class II$^{neg}$ population (Fig 3E and 3F), which has high phagocytic activity [38]. It will be important to clarify if the differences in the distribution of macrophage populations can be used as biomarkers to stratify asthma patients.

Analyzing T cells, we noted the increase of Th17 cells and IL-17 production in neutrophilic asthma (Fig 4A and 4B), which is in line with previous reports [82]. In contrast to the Th17 cells, however, the frequency of NKT17 cells was decreased in neutrophilic asthma (Fig 4D), suggesting that *i*NKT cells might be less relevant for the pathology of this asthma endotype.

iBALTs (inducible bronchus-associated lymphoid tissues) are tertiary lymphoid tissue, structurally similar to germinal centers in the lymph nodes, which are formed in the airways during lung inflammation, for example, following infections or during chronic diseases, like COPD (chronic obstructive pulmonary disease) [52]. In allergic asthma models, iBALTs support the Th2 cell responses induced by fungal infection [83] or LPS [84]. Within iBALTs, germinal center (GC) B cells can act as APCs by presenting inhaled antigens to T cells and can drive their differentiation into Th2 cells [50]. Conversely, antigen-specific Th2 cells can support iBALT formation [85]. However, the role of iBALT in neutrophilic asthma and how it would compare to eosinophilic asthma is not known. We provide here, to our knowledge, the first direct comparison of GC B cells and iBALTs in these two asthma endotypes. Histological examination did not indicate that the size of the iBALTs differed between the two endotypes (Fig 5D). However, we found that in neutrophilic but not eosinophilic asthma the frequency of GC B cells was greatly increased (Fig 5C) and that they produce more IFNγ following activation (Fig 5B). These data suggest that activated B cells, locally in the inflamed lung, might contribute to the pathology in neutrophilic asthma.

Our side-by-side comparison of BALF and lung tissue also revealed several unexpected discrepancies. Although we observed clear differences between eosinophilic and neutrophilic asthma in the lungs for basophils, mast cells, cDCs, CD103$^{+}$ DCs (Fig 2B–2E), as well IMs (Fig 3B), these changes were not represented in the BALF. As BALF of asthma patients is easier accessible than lung tissue, it is important to know which cell populations in the BALF are most indicative for the lung inflammation. Our data indicate that both pDCs (Fig 2F) and ExMs (Fig 3C) were markedly increased in the BALF of mice with neutrophilic but not eosinophilic asthma, making them possible candidates for the clinical diagnosis of asthma endotypes.

In summary, we report here, to our knowledge for the first time, a direct side-by-side comparison of the main myeloid cell populations, *i*NKT cells, and GC B cells from the inflamed lungs of mice with adjuvant-induced eosinophilic and neutrophilic asthma. These data suggest that the subset distribution of macrophages and dendritic cells in the BALF could be used to aid the determination of asthma endotypes. Although further research is required to verify these results in asthma patients, the differential distribution of myeloid cells, other than eosinophils and neutrophils, promises to be helpful as an early biomarker to support the stratification of asthma endotypes.

## Material and methods

### Mice

All mice were housed in the vivarium of the Izmir Biomedicine and Genome Center (IBG, Izmir, Turkey) in accordance with the respective institutional animal care committee guidelines. All mouse experiments were performed with prior approval by the institutional ethic committee ('Izmir Biomedicine and Genome Center's Ethical Committee on Animal

Experimentation'), in accordance with national laws and policies. All the methods were carried out in accordance with the approved guidelines and regulations.

## Reagents, monoclonal antibodies, and flow cytometry

Monoclonal antibodies against the following mouse antigens were used in this study: CD3ε (145.2C11, 17A2), CD4 (RM4-5), CD8α (53–6.7, 5H10), CD11b (M1/70), CD11c (N418), CD19 (1D3, 6D5), CD24 (M1/69), CD44 (IM7), CD45 (30-F11), CD45.2 (104) CD45R/B220 (RA3-6B2), CD64 (X54-5/7.1), CD95 (SA367H8), CD103 (3E7), CD117/cKit (2B8), CD122 (TM-beta1), CD127 (A7R34, SB/199), CD170/Siglec F (E50-2440), F4/80 (BM8), FcεRIα (MAR-1), FoxP3 (FJK-16s), Gata3 (L50-823), GL7 (GL7), IFNγ (XMG1.2), IL-4 (11B11), IL-13 (13A), IL-17A (TC11-18H10), Ly6G (1A8), Ly6C (HK1.4), MHC class II (M5/114.15.2), NK1.1 (PK136), PLZF (9E12), RORγt (Q31-378), Tbet (O4-46), TCRβ (H57-597), TNF (MP6-XT22). Antibodies were purchased from BD Biosciences (San Diego, CA), BioLegend (San Diego, CA), eBioscience (San Diego, CA), or ThermoFisher Scientific (Carlsbad, CA). Antibodies were biotinylated or conjugated to Pacific Blue, eFluor 450, Brilliant Violet 421, V500, Brilliant Violet 510, Brilliant Violet 570, Brilliant Violet 650, Brilliant Violet 711, Brilliant Violet 785, Brilliant Violet 786, FITC, Alexa Fluor 488, PerCP-Cy5.5, PerCP-eFluor 710, PE, PE-CF594, PE-Cy7, APC, Alexa Fluor 647, eFluor 660, Alexa Fluor 700, APC-Cy7, APC-eFluor 780 or APC-Fire750. Anti-mouse CD16/32 antibody (2.4G2) used for Fc receptor blocking was obtained from Tonbo Biosciences. Unconjugated mouse and rat IgG antibodies were purchase from Jackson ImmunoResearch (West Grove, PA). Dead cells were labeled with Zombie UV Dead Cell Staining Kit (BioLegend). Flow cytometry was performed as described [86]. Fluorochrome-conjugated CD1d tetramers were either were obtained from the NIH tetramer core facility (Emory University, Atlanta, GA) or prepared as described previously [87]. Graphs derived from digital data are displayed using a 'bi-exponential display' [88]. The gating strategy utilized is outlined in S2 and S3 Figs.

## Adjuvant-induced asthma models

Neutrophilic asthma was induced by one injection (d0) per mouse with 0.5 mg/mL CFA (Complete Freund's Adjuvant, Sigma-Aldrich, St. Louis, MO) mixed with 20 μg OVA (Ovalbumin, Hyglos, Germany) [15, 21]. Eosinophilic asthma was induced by weekly injections (d0, d7, d14) per mouse of 1 mg Alum ('Imject Alum', ThermoFisher Scientific) mixed with 20 μg OVA [15, 21]. Seven days after the last Alum/OVA injection, all mice were challenged with 50 μg OVA/mouse by pharyngeal/laryngeal installation once on two consecutive days (d21, d22). 16–18 hours later, the mice were sacrificed, and the blood, BALFs, and lungs were collected to assess immune responses by flow cytometry, ELISA, and immunohistochemistry/immunofluorescence.

## Cell preparation

Bronchoalveolar lavage fluid (BALF) was collected by inflating the murine lungs with 1 mL ice-cold PBS, which was repeated twice with fresh PBS. The three washes were centrifuged separately at 400 *g* for 7 min at 4˚C. The supernatant of the first wash was collected for the ELISA analysis, the cells of all three washes were pooled for the flow cytometric analysis. Single-cell suspensions from mouse lungs were prepared as described [89]. In brief, lungs were removed and minced into smaller pieces in a 6-well plate (Greiner, Germany). The digestion mixture, composed of 1 mg/mL collagenase D and 0.1 mg/mL DNase I (both from Roche, Switzerland) in complete RPMI medium (RPMI 1640 medium (Life Technologies); supplemented with 10% (v/v) fetal calf serum (FCS), 1% (v/v) Pen-Strep-Glutamine (10.000 U/ml

penicillin, 10.000 μg/ml streptomycin, 29.2 mg/ml L-glutamine (Life Technologies)) and 50 μM β-mercaptoethanol (Sigma)), was added to the samples and incubated for 30 min at 37˚C on a lateral shaker. The lung samples were filtered through 100 μm mesh with PBS, washed twice, and the red blood cells were eliminated by ACK lysis buffer (Lonza, US).

## ELISA

The IFNγ, IL-5, and IL-13 cytokine levels in BALF were measured with the respective Sandwich-ELISA kits (R&D Systems, MN, US) according to the manufacturer's instructions. OVA-specific antibodies were measured in the sera, collected from mice via cardiac puncture, as described [90]. For detecting total IgG, IgG1, IgG2, and IgA antibodies, the sera were serial diluted and loaded onto 10 μg OVA-coated plates. Horse-radish peroxidase (HRP)-conjugated anti-mouse IgG, IgG1, IgG2, IgA (Southern Biotech, USA) antibodies were used as detection antibodies. For the detection of OVA-specific IgE antibodies, the sera were diluted 1/10 with PBS containing 1% (w/v) BSA and loaded onto anti-mouse IgE coated (BD biosciences) plates anti-OVA-HRP antibodies (AbD Serotec, BioRad) were used as detection antibody [90]. The colorimetric change, resulting from the enzymatic reaction between the HRP portion of detection antibody and the substrate TMB, was measured as absorbance at 450 nm (OD450) by Spectrophotometer (Thermo Scientific, Multiskan FC Microplate Photometer). The titers were defined from the reciprocal value of the absorbance at OD450.

## In vitro stimulation

Lung-derived lymphocytes were stimulated *in vitro* with PMA (50 ng/mL) and ionomycin (1 μg/mL) (both Sigma-Aldrich, St. Louis, MO) for four hours at 37˚C in the presence of both Brefeldin A (GolgiPlug) and Monensin (GolgiStop). As GolgiPlug and GolgiStop (both BD Biosciences, San Diego, CA) were used together, half the amount recommended by the manufacturer were used, as suggested previously [91].

## Histology

Lungs were inflated with 4% PFA (Cell Signaling, San Diego, US) and fixed for four hours on a lateral shaker at 4˚C. After dehydration with 30% sucrose overnight on a lateral shaker at 4˚C, samples were embedded in O.C.T (Tissue Tek, Sakura, US) and snap-frozen. 7–10 μm thick sections were prepared with a cryostat (Leica CM 1950). For the H&E staining, the tissue sections were stained with hematoxylin (Sigma, USA) and eosin (Sigma, USA) for two minutes each. Slide contrast was increased by a brief HCl/ethanol treatment (1/1000, v/v) (Sigma, USA). Slides were fixed by ascending concentration of ethanol (70%, 80%, 90%, 95%, 100%) (Sigma, USA) and a final 5 second xylene (Sigma, USA) treatment. For the Alcian Blue/PAS staining, the tissue sections were stained with an Alcian Blue/PAS staining kit according to the manufacturer's recommendations (Bio Optica, Italy). Once the slides dried, they were mounted with entellan and examined with a light microscope (Olympus IX71). For the immunofluorescence, tissue sections were stained with CD45R-AF647 (BD Biosciences, San Diego, CA, US) and DAPI (ThermoFisher Scientific, Carlsbad, CA) and analyzed with a confocal microscope (Zeiss LSM 880).

## Statistical analysis

Data are presented as mean ± standard error of the mean (SEM). The statistical analysis was performed with GraphPad Prism 7.0 software (GraphPad Software, San Diego, CA). One-way

ANOVA followed by Holm-Sidak posthoc test are used to compare p values regarded as $^*p \leq 0.05$, $^{**}p \leq 0.01$, and $^{***}p \leq 0.001$.

## Supporting information

**S1 Fig. Supporting data to Fig 1.** C57BL/6 mice were immunized as outlined in Fig 1A to induce neutrophilic (CFA/OVA) or eosinophilic (Alum/OVA) asthma. **(A, B)** Cells from bronchoalveolar lavage fluid (BALF) were stained for (A) eosinophils (live CD45$^+$ CD19$^-$ CD11c$^-$ CD11b$^-$ Ly6G$^-$ Siglec-F$^+$) and (B) neutrophils (live CD45$^+$ CD19$^-$ CD11b$^{+/lo}$ Ly6G$^+$). **(C)** Serum levels of OVA-specific IgG2b and IgG2c antibodies in the indicated groups were determined by ELISA. Combined data from four independent experiments are shown (naïve group n = 6 (C) or 12 (A, B) mice/group, experimental asthma groups n = 16 (C) or 17–18 (A, B) mice/group in total).
(TIF)

**S2 Fig. Gating strategy to identify myeloid cells in the inflamed lung.** A graphic outline **(A)** and exemplary graphs **(B)** are given to illustrate the gating strategy employed to identify myeloid cells in the lung and BALF. Alveolar macrophages (AMs): live CD45$^+$ CD19$^-$ Siglec-F$^+$ Ly6G$^-$ CD11c$^+$ cells; Basophils: live CD45$^+$ CD19$^-$ Siglec-F$^-$ Ly6G$^-$ CD11b$^+$ FcɛRIα$^+$ CD117$^-$ cells; Conventional DCs (cDC): live CD45$^+$ CD19$^-$ Siglec-F$^-$ Ly6G$^-$ CD24$^+$ CD11c$^+$ MHC class II$^+$ CD103$^-$ CD11b$^+$ cells; CD103$^+$ DCs: live CD45$^+$ CD19$^-$ Siglec-F$^-$ Ly6G$^-$ CD24$^+$ CD11c$^+$ MHC class II$^+$ CD103$^+$ cells; Dendritic cells (all): live CD19$^-$ CD45$^+$ Siglec-F$^-$ Ly6G$^-$ F4/80$^-$ CD64$^-$ CD24$^+$ CD11c$^+$ MHC class II$^{+/-}$; Eosinophils: live CD45$^+$ CD19$^-$ CD11c$^-$ CD11b$^-$ Ly6G$^-$ Siglec-F$^+$ cells; Exudate macrophages (ExMs): live CD45$^+$ CD19$^-$ Siglec-F$^-$ Ly6G$^-$ CD24$^-$ F4/80$^+$ CD64$^{+/-}$ Ly6C$^+$ CD11b$^+$ cells; Interstitial macrophages (IMs): live CD45$^+$ CD19$^-$ Siglec-F$^-$ Ly6G$^-$ CD24$^-$ F4/80$^+$ CD64$^{+/-}$ Ly6C$^+$ CD11b$^-$ cells; Neutrophils: live CD45$^+$ CD19$^-$ CD11b$^{+/lo}$ Ly6G$^+$ cells; Macrophages (all): live CD19$^-$ CD45$^+$ Siglec-F$^-$ Ly6G$^-$ F4/80$^+$ CD64$^+$ cells; Mast cells: live CD45$^+$ CD19$^-$ Siglec-F$^-$ Ly6G$^-$ CD11b$^+$ FcɛRIα$^+$ CD117$^+$ cells; Plasmacytoid dendritic cells: pDC, live CD19$^-$ CD45$^+$ Siglec-F$^-$ Ly6G$^-$ CD11b$^-$ CD45R$^+$ Ly6C$^+$ cells.
(TIF)

**S3 Fig. Gating strategy to identify lymphoid cells in the inflamed lung.** A graphic outline **(A)** and exemplary graphs **(B)** are given to illustrate the gating strategy employed to identify lymphoid cells in the lung and BALF. Th1 cells (live CD19/CD45R$^-$ CD3ɛ$^+$ CD4$^+$ Tbet$^+$ cells), Th2 cells (live CD19/B220$^-$ CD3ɛ$^+$ CD4$^+$ Gata3$^+$ cells), Th17 cells (live CD19/CD45R$^-$ CD3ɛ$^+$ CD4$^+$ RORγt$^+$ cells), and Tregs (live CD19/CD45R$^-$ CD3ɛ$^+$ CD4$^+$ CD127$^{lo/-}$ FoxP3$^+$), $i$NKT cells (live CD19/CD45R$^-$ CD3ɛ$^+$ CD1d/PBS57-tetramer$^+$ cells) and its subsets in the lung NKT1 (PLZF$^{lo}$ RORγt$^-$), NKT2 cells (PLZF$^{int/hi}$ RORγt$^-$), NKT17 cells (PLZF$^{int}$ RORγt$^+$) cells are shown.
(TIF)

## Acknowledgments

The authors wish to thank the Flow Cytometry Core Facility, Histopathology Core Facility, Imaging Core Facility, and the vivarium at the Izmir Biomedicine and Genome Center (IBG) for technical assistance. We are grateful to the NIH Tetramer Core Facility for providing the PBS57-loaded mouse CD1d tetramers.

## Author Contributions

**Conceptualization:** Müge Özkan, Gerhard Wingender.

**Data curation:** Müge Özkan, Yusuf Cem Eskiocak, Gerhard Wingender.

**Formal analysis:** Müge Özkan.

**Funding acquisition:** Gerhard Wingender.

**Investigation:** Müge Özkan, Yusuf Cem Eskiocak.

**Methodology:** Müge Özkan, Gerhard Wingender.

**Project administration:** Gerhard Wingender.

**Resources:** Gerhard Wingender.

**Supervision:** Gerhard Wingender.

**Validation:** Müge Özkan, Yusuf Cem Eskiocak.

**Visualization:** Müge Özkan, Gerhard Wingender.

**Writing – original draft:** Müge Özkan, Yusuf Cem Eskiocak, Gerhard Wingender.

**Writing – review & editing:** Müge Özkan, Gerhard Wingender.

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
