## [Decision Letter · Decision Letter 0]

29 Apr 2021

PONE-D-21-10740

Macrophage and dendritic cell subset composition can distinguish endotypes in adjuvant-induced asthma mouse models

PLOS ONE

Dear Dr. Wingender,

Thank you for submitting your manuscript to PLOS ONE. After careful consideration, we feel that it has merit but does not fully meet PLOS ONE’s publication criteria as it currently stands. Therefore, we invite you to submit a revised version of the manuscript that addresses the points raised during the review process.

We look forward to receiving your revised manuscript.

Kind regards,

Amarjit Mishra, PhD

Academic Editor

PLOS ONE

Journal Requirements:

Reviewers' comments:

Reviewer's Responses to Questions

**Comments to the Author**

1. Is the manuscript technically sound, and do the data support the conclusions?

Reviewer #1: Yes

2. Has the statistical analysis been performed appropriately and rigorously? 

Reviewer #1: Yes

3. Have the authors made all data underlying the findings in their manuscript fully available?

Reviewer #1: Yes

4. Is the manuscript presented in an intelligible fashion and written in standard English?

Reviewer #1: Yes

5. Review Comments to the Author

Reviewer #1: Comments for the Author:

In the manuscript authors have used eosinophilic and neutrophilic inflammation of murine model and intends to characterize their immune infiltrate to corresponding endotypes. More specifically, author characterized eosinophilic inflammation by preferential recruitment of interstitial macrophages and myeloid dendritic cells, whereas neutrophilic asthma enriches plasmacytoid dendritic cells, exudate macrophages, and GL7 + activated B cells. Author claims that differential distribution of macrophage and dendritic cell subsets could be used as biomarkers to diagnose asthma endotypes.

This study adds an incremental notion about various immune cells involved during the airway’s inflammation specifically eosinophilic and neutrophilic which is very interesting and relevant to the field. However, there are few concerns regarding experiments, figures, and data. My specific comments are:

Comments:

1. Figure1C, naïve animal its self-shows 10% to 20% neutrophils in BALF and Lung, respectively. Can author explain why?

2. Allergen specific mediastinal lymph node re-stimulation and cytokine (Th1/Th2/Th17) estimation in supernatant provides more appropriate idea to conclude the immune response generated during inflammation. Fig1E, the cytokine was measured in BALF samples which might add more variability in data. How was the data normalized?

3. Author should also include the chemokines which are majorly secreted during inflammation either eosinophilic or neutrophilic. ELISA data showing IgG2b or IgG2b could be removed from figure 1.

4. Figure4B, author differentiated the Th1/Th2/Th17 cell and NKT1/NKT2/NKT3 cell using combination of cell specific surface markers and transcription factors. At same time author have measured the cytokine in the lung samples. How will author explain about the source of cytokines as Th cells as well as NKT cells express T cell markers like CD3 or CD4?

5. During lung cell preparation complete RPMI media was used. What is meaning of “complete RPMI media”?

6. Figure1D and Figure3D with poor quality.

6. PLOS authors have the option to publish the peer review history of their article (what does this mean?). If published, this will include your full peer review and any attached files.

Reviewer #1: **Yes: **Anil Kumar Jaiswal

---

## [Author Response · Author response to Decision Letter 0]

5 May 2021

Please refer to the point-to-point reply.

---

## [Decision Letter · Decision Letter 1]

19 May 2021

Macrophage and dendritic cell subset composition can distinguish endotypes in adjuvant-induced asthma mouse models

PONE-D-21-10740R1

Dear Dr. Wingender,

We’re pleased to inform you that your manuscript has been judged scientifically suitable for publication and will be formally accepted for publication once it meets all outstanding technical requirements.

Kind regards,

Amarjit Mishra, PhD

Academic Editor

PLOS ONE

Reviewers' comments:

Reviewer's Responses to Questions

**Comments to the Author**

1. If the authors have adequately addressed your comments raised in a previous round of review and you feel that this manuscript is now acceptable for publication, you may indicate that here to bypass the “Comments to the Author” section, enter your conflict of interest statement in the “Confidential to Editor” section, and submit your "Accept" recommendation.

Reviewer #1: All comments have been addressed

2. Is the manuscript technically sound, and do the data support the conclusions?

Reviewer #1: Yes

3. Has the statistical analysis been performed appropriately and rigorously? 

Reviewer #1: Yes

4. Have the authors made all data underlying the findings in their manuscript fully available?

Reviewer #1: Yes

5. Is the manuscript presented in an intelligible fashion and written in standard English?

Reviewer #1: Yes

6. Review Comments to the Author

Reviewer #1: Comments for the Author:

The manuscript demonstrates eosinophilic and neutrophilic inflammation of murine model and intends to characterize their immune infiltrate to corresponding endotypes. More specifically, author characterized eosinophilic inflammation by preferential recruitment of myeloid dendritic cells, whereas neutrophilic asthma enriches plasmacytoid dendritic cells and GL7 + activated B cells. Author claims that differential distribution of macrophage and dendritic cell subsets could be used as biomarkers to diagnose asthma endotypes.

All the query raised has been addressed by author. The revised manuscript is quite improved, and it could be accepted.

7. PLOS authors have the option to publish the peer review history of their article (what does this mean?). If published, this will include your full peer review and any attached files.

Reviewer #1: **Yes: **Anil Kumar Jaiswal

---

## [Editor Report · Acceptance letter]

21 May 2021

PONE-D-21-10740R1 

Macrophage and dendritic cell subset composition can distinguish endotypes in adjuvant-induced asthma mouse models 

Dear Dr. Wingender:

I'm pleased to inform you that your manuscript has been deemed suitable for publication in PLOS ONE. Congratulations! Your manuscript is now with our production department. 

Kind regards, 

on behalf of

Dr. Amarjit Mishra 

Academic Editor

PLOS ONE